# The Effect of Precipitation pH on Protein Recovery Yield and Emulsifying Properties in the Extraction of Protein from Cold-Pressed Rapeseed Press Cake

**DOI:** 10.3390/molecules27092957

**Published:** 2022-05-05

**Authors:** Cecilia Ahlström, Johan Thuvander, Marilyn Rayner, María Matos, Gemma Gutiérrez, Karolina Östbring

**Affiliations:** 1Department of Food Technology Engineering and Nutrition, Lund University, Naturvetarvägen 12, 223 62 Lund, Sweden; johan.thuvander@food.lth.se (J.T.); marilyn.rayner@food.lth.se (M.R.); karolina.ostbring@food.lth.se (K.Ö.); 2Department of Chemical and Environmental Engineering, University of Oviedo, Julián Clavería 8, 33006 Oviedo, Spain; matosmaria@uniovi.es (M.M.); gutierrezgemma@uniovi.es (G.G.)

**Keywords:** rapeseed press cake, protein extraction, protein recovery, plant proteins, emulsion stability

## Abstract

Rapeseed is the second most cultivated oilseed after soybean and is mainly used to produce vegetable oil. The by-product rapeseed press cake is rich in high-quality proteins, thus having the possibility of becoming a new plant protein food source. This study aimed to investigate how the precipitation pH affects the protein yield, protein content, and emulsifying properties when industrially cold-pressed rapeseed press cake is used as the starting material. Proteins were extracted under alkaline conditions (pH 10.5) with an extraction coefficient of 52 ± 2% followed by precipitation at various pH (3.0–6.5). The most preferred condition in terms of process efficiency was pH 4.0, which is reflected in the zeta potential results, where the proteins’ net charge was 0 at pH 4.2. pH 4.0 also exhibited the highest protein recovery yield (33 ± 0%) and the highest protein concentration (64 ± 1%, dry basis). Proteins precipitated at pH 6.0–6.5 stabilized emulsions with the smallest initial droplet size, although emulsions stabilized by rapeseed protein precipitated at pH 5.0–6.0 showed the highest emulsion stability at 37 °C for 21 days, with a limited layer of free oil. Overall, emulsion stabilized by protein precipitated at pH 5.0 was the most stable formulation, with no layer of free oil after 21 days of incubation.

## 1. Introduction

The interest in plant-based food is growing, both from consumers and from the food industry [1]. The agricultural sector is responsible for around 30% of greenhouse gas emissions [2] and one strategy to reduce these emissions is to upcycle underutilized agricultural side-streams. FAO has listed increased agricultural productivity and decreased food waste as priority actions to fight climate change and food insecurity [3]. The high-value utilization of by-products in the agro-food industry can increase economic value and sustainability [4,5]. Examples of approaches taken to valorize side-streams are the use of proteins from leafy green residues from broccoli and kale for food purposes [6] and the use of protein nanofibers from whey protein isolate to produce a coating system to prolong shelf life and prevent deterioration during storage of food products [7]. Residues from apple juice production (apple pomace) have been used as a substrate for mushroom production [8] and coffee grounds extract has been used as a component in food packaging systems [9]. Another underused agricultural by-stream is the press cake from rapeseed oil production.

Rapeseed (*Brassica napus* L., *Brassica rapa*, and *Brassica juncea* of rapeseed quality) is the second most commonly cultivated oilseed after soybean [10] and in 2020, 72 million tons of rapeseed were harvested globally [11]. The oil is mainly used as a high-quality edible oil but can also be used in the production of biodiesel and bioplastics [12]. During the oil pressing process, a protein-rich press cake (36–40% protein, dry basis) is co-produced and the ratio of oil to press cake is 1:2 [13]. Rapeseed protein exhibits a balanced amino acid profile comparable with soy protein and is therefore suggested as an alternative dietary protein source [14]. Despite the high protein quality, the press cake is mainly used as animal feed and fertilizer [12] due to the presence of antinutritional compounds, such as glucosinolates, phenolic compounds, and phytic acid [15]. Therefore, rapeseed proteins need to be purified before being used as an ingredient in food products [13].

Oil can be extracted from rapeseed using different processing methods, with the most common being hot-pressing. This process involves a cooking step (80–105 °C) before the seeds are pressed through a series of screw presses to extract the oil. The hot-pressed meal usually contains about 4% oil, and solvents (e.g., hexane) can be used to extract the remaining oil [16]. Rapeseed oil can also be produced under milder conditions, i.e., cold pressing. The process starts with cleaning of the seeds and adjustment of the moisture content, whereafter the seeds are placed in a screw press, where the oil is extracted. The temperature of the oil usually does not exceed 40 °C and the press cake contains 16–20% oil after pressing. Studies have shown that rapeseed proteins exposed to heat during processing, e.g., hot-pressing, exhibit lower protein extractability and compromised technological functionality compared to proteins processed under milder conditions [10,17].

One of the most common methods used to isolate proteins from rapeseed press cake is the pH-shift method. The press cake is first milled and soaked in an alkaline solution in which the proteins are solubilized [14,18]. The extraction coefficient (or extraction yield as it is also called) typically varies between 40% and 70%, where a higher extraction pH is associated with a higher extraction coefficient but also comprised functional and organoleptic properties [19]. After the extraction, the slurry is separated into a spent solids fraction containing husks, intact cells, and non-solubilized protein, and a light liquid phase containing solubilized protein and co-extracted carbohydrates. The pH is then adjusted to the proteins isoelectric point (pI) and the proteins are subjected to precipitation. In the following separation step, the precipitated proteins are recovered [14,20].

Rapeseed has a complex protein composition, where the two most abundant proteins are the storage proteins cruciferin (12 S globulin) and napin (2 S albumin) [12]. Cruciferin accounts for 40–60% of the proteins in the seed and is a neutral protein that is soluble in salt solutions, and has a high molecular weight (300–350 kDa) [13,21]. Cruciferin has six subunits (50 kDa) that form a hexametric structure [22], where each subunit consists of an α-chain (29–33 kDa) and a β-chain (20–23 kDa) linked by a disulfide bond [23]. The pI of cruciferin is 7.2 [24]. The other major storage protein in rapeseed is napin, which accounts for around 20% of the total protein content in the seed and is a water-soluble alkaline protein with a molecular weight of 12–16 kDa [13,21]. Napin consists of two subunits, a large polypeptide chain (9 kDa) and a small polypeptide chain (4 kDa) linked together by a disulfide bond [25]. Napin has a pI of around 11 [26]. Other important minor proteins in rapeseed are the oil body proteins (OBPs), of which oleosin (16–18 kDa) is the dominating species followed by caleosin (27 kDa) and steroleosin (39–41 kDa) [13,27]. Oleosin is an alkaline protein that accounts for 8–20% of the total protein content [28]. The oleosin protein structure consists of three domains: two terminal amphipathic regions and a central hydrophobic region [27]. The pI for OBPs is 6.5 [29]. The pI for the complex mixture of rapeseed proteins has been reported to be in the pH range of 3.5–6 and botanical variations are suggested to contribute to the wide range of pI values reported in the literature [30].

The three major proteins in rapeseed, cruciferin, napin, and oleosin, differ in their amino acid sequence, resulting in major differences in secondary, tertiary, and quaternary structures. These structural differences influence the proteins’ techno-functionality, such as emulsifying properties, and therefore it is interesting to separate the different species to amplify the function in food applications [24].

From the literature, it is not clear which kind of protein in rapeseed demonstrates the best emulsifying capacity and stability. Some authors reported napin to be a more efficient emulsifier compared to cruciferin due to its smaller size and napin was therefore suggested to have a higher diffusion rate compared to the larger cruciferin molecule [31,32]. Others reported that cruciferin stabilizes smaller emulsion droplets with higher stability compared to napin, with a superior emulsifying activity index (EAI) [12,33]. The conflicting results can be attributed to the different extraction methods and different emulsion formulations. The literature is scarce regarding emulsion studies that have investigated the emulsifying properties of a wide range of precipitation pH values at several protein concentrations, the latter of which is important for identifying the concentration at which the oil–water interface is fully covered with proteins. Moreover, there is a lack of data on the emulsion stability over time for emulsions prepared under pH conditions reflecting commercial food emulsions.

In the present study, we isolated rapeseed protein from cold-pressed rapeseed press cake and varied the precipitation pH to investigate possible effects on the protein recovery yield, and the corresponding emulsifying properties and emulsion stability of the protein concentrates using the Turbiscan method. A lab-on-chip method was also used for the quantitative characterization of the proteins.

## 2. Materials and Methods

### 2.1. Raw Materials and Chemicals

Cold-pressed rapeseed press cake (RPC) (*Brassica napus* L.) was a kind gift from Gunnarshögs Jordbruk AB (Hammenhög, Sweden) and was used in all trials. No solvents were used during the oil extraction and the oil temperature did not exceed 35 °C. The dry matter of the rapeseed press cake was 91% (AOAC 2007) and the protein content on a dry basis was 31% using the Dumas method. The RPC was stored at −18 °C before the onset of the trials.

Citric acid, DL-Dithiothreitol (DTT), (S)-2-Aminobutane-1,4-dithiol hydrochloride (DTBA), and sodium hydroxide were purchased from Merck (Darmstadt, Germany). The protein 230 kit for the protein profile analysis, including all reagents, was purchased from Agilent Technologies (Agilent, Santa Clara, CA, USA). Miglyol 812 was purchased from Sasol (Witten, Germany). De-oiled lecithin from soy was purchased from Cargill (Minneapolis, MN, USA). All other chemicals were of analytical grade.

### 2.2. Protein Extraction and Recovery

Proteins were recovered from RPC (Figure 1) as previously described by Östbring et al. [34] and the method was modified from Wijesundera et al. [29]. The RPC (50 g) was ground using a knife mill (Grindomix GM 200, Retsch, Germany) for 20 s and then was dispersed in tap water (1:9 *w*/*w*). The pH was adjusted to 10.5 with 2 M NaOH and the dispersion was stirred at 750 rpm using a Rushton impeller with a diameter of 30 mm (IKA Labortechnik, Eurostar digital, Staufen, Germany). The pH was readjusted to 10.5 after 10 min followed by incubation for a total of 4 h under continuous stirring. The dispersion was separated by centrifugation (Beckman Coulter, Avanti^®^J-15R Centrifuge, Brea, CA, USA) at 4700× *g*, 20 °C for 20 min. The separated solids mainly containing seed coating and fibers, hereafter referred to as the spent solids, were discarded. The supernatant, hereafter referred to as the light liquid phase, was collected. The extraction coefficient was defined as:(1)$$\mathrm{Extraction}\;\mathrm{coefficient}=\frac{\mathrm{mass}\;\mathrm{of}\;\mathrm{protein}\;\mathrm{in}\;\mathrm{the}\;\mathrm{light}\;\mathrm{liquid}\;\mathrm{phase}\;\left(g\right)\;}{\mathrm{mass}\;\mathrm{of}\;\mathrm{protein}\;\mathrm{in}\;\mathrm{the}\;\mathrm{rapeseed}\;\mathrm{press}\;\mathrm{cake}\;\left(g\right)}$$
and is a measurement of how large a proportion of the protein could be solubilized during the extraction step. Precipitation of the proteins was carried out at 8 different acidic pH ranging from 3.0 to 6.5 in half pH increments. The pH of the light liquid phase was adjusted with citric acid, and solids were separated by centrifugation as described above. The precipitate and the supernatant fraction, hereafter referred to as the light phase with unprecipitated proteins, were collected for further analysis. The precipitation coefficient was defined as:(2)$$\mathrm{Precipitation}\;\mathrm{coefficient}=\frac{\mathrm{mass}\;\mathrm{of}\;\mathrm{protein}\;\mathrm{in}\;\mathrm{the}\;\mathrm{precipitate}\;\left(g\right)\;}{\mathrm{mass}\;\mathrm{of}\;\mathrm{protein}\;\mathrm{in}\;\mathrm{the}\;\mathrm{light}\;\mathrm{liquid}\;\mathrm{phase}\;\left(g\right)}$$
and is a measurement of how large a proportion of the protein was available in the light liquid phase that was precipitated at a specific pH. Protein recovery yield was defined as:(3)$$\mathrm{Protein}\;\mathrm{recovery}\;\mathrm{yield}=\frac{\mathrm{mass}\;\mathrm{of}\;\mathrm{protein}\;\mathrm{in}\;\mathrm{the}\;\mathrm{precipitate}\;\left(g\right)\;}{\mathrm{mass}\;\mathrm{of}\;\mathrm{protein}\;\mathrm{in}\;\mathrm{the}\;\mathrm{rapeseed}\;\mathrm{press}\;\mathrm{cake}\;\left(g\right)}$$
and is a measurement of how large a proportion of the protein in the RPC was obtained in the rapeseed precipitate fraction.

The subsequent precipitates were freeze-dried using a laboratory freeze dryer (Hetosicc freeze dryer CD 12, Birkerod, Denmark). The samples were distributed into aluminum trays to form a layer of a maximum of 10 mm and thereafter frozen at −18 °C for 24 h before freeze-drying. The plate temperature was 20 °C, the condenser was −50 °C, and the vacuum pressure of the dryer was 0.02 mbar. The residence time for the samples in the freeze-dryer was seven days. After termination of the freeze-drying, the samples were placed in a desiccator for two days to remove any remaining moisture. The resulting powder was stored in the freezer (−18 °C) until evaluation. The other collected samples of the other phases from the extraction process were stored in a freezer at −18 °C before analysis. The rapeseed protein isolation process was performed in triplicates for each acidic pH.

### 2.3. Proximate Analysis

#### 2.3.1. Dry Matter Content

The dry matter content of the RPC, spent solids, and protein precipitates was analyzed according to the official method of analysis (AOAC 2007). Briefly, samples were placed in a convective oven (Termaks, Bergen, Norway) at 103 °C until constant weight (>16 h). The analysis was performed in duplicates for each batch (*n* = 6).

#### 2.3.2. Protein Concentration

The protein concentration of the RPC, spent solids, and precipitates was determined using the Dumas method (Flash EA, 1112 Series, Thermo Electron Co., Waltham, MA, USA). Approximately 25–50 mg of sample were placed inside a tin disc (33 mm) for analysis, air was used as the blank, and aspartic acid was used as the reference. A nitrogen-to-protein conversion factor of 6.25 was used for all analyses. The analysis was performed in duplicates on each batch (*n* = 6).

### 2.4. Zeta Potential

The zeta potential can be used to identify the pI of a protein solution. The zeta potential was analyzed with a Malvern Zetasizer Nano ZS (Malvern Instruments Ltd., Worcestershire, UK) at 25 °C and the pH of the light liquid phase was adjusted to pH in the interval of 3.0–6.5 with citric acid. The dispersions were diluted 10 times in deionized water prior to analysis and duplicate measurements on each batch (*n* = 6) with 3 consecutive runs were performed.

### 2.5. Protein Profile Analysis

A lab-on-a-chip capillary electrophoresis method was used to identify the protein species [35]. The Agilent Bioanalyzer 2100 (Agilent, Santa Clara, CA, USA) was used to examine the protein profile and estimate the molecular weights of the protein. Precipitated proteins can be difficult to re-disperse in solutions and therefore, the proteins that remained dissolved in the light liquid phases and light phases with unprecipitated proteins originating from each acidic pH were analyzed to provide a more reliable result. Investigation of the unprecipitated proteins provides an overview of the proteins’ precipitation at different pH. DL-Dithiothreitol (DTT) was used as a reducing agent for the light liquid phases and light phases with unprecipitated proteins with a pH of 5.0–6.5. For light phases with unprecipitated proteins with a pH of 3.0–4.5, (S)-2-Aminobutane-1,4-dithiol hydrochloride (DTBA) was used instead of DTT due to its capacity to separate proteins at low pH. The reducing agent was incorporated in the included regents. Cruciferin is the predominant large-molecular-weight protein and, therefore, a kit suitable for proteins between 14 and 230 kDa was used. All samples were prepared according to the manufacturer’s instructions in at least triplicates for each batch (*n* = 9) and the Bioanalyzer software was used for calculations.

### 2.6. Emulsion Formulation and Particle Size Analysis

In order to determine how the precipitation pH affected the emulsifying properties of the rapeseed precipitates, an emulsion series was performed. Oil-in-water emulsions were prepared in glass tubes with 2.33 mL of dispersed phase (Miglyol 812) and 4.67 mL of continuous phase (phosphate-buffered saline: 0.01 M phosphate, 0.0027 M KCl, 0.137 M NaCl, pH 7.4). The emulsions had a pH of 4.8 ± 0.1, which was chosen to reflect the pH of mayonnaise. Benzoic acid (0.1%) was added before emulsification to avoid microbial growth during storage and the addition did not affect the emulsifying properties. Freeze-dried rapeseed precipitates precipitated at various acidic pH were added and emulsion series with a protein concentration of 0.5, 1, 2.5, 5, 10, and 50 mg rapeseed protein/mL oil were formulated. Soy lecithin was used as a control in the same concentrations as rapeseed protein. The emulsions were homogenized with a high-performance rotor-stator homogenizer (Silent Crusher M, Heidolph Instruments GmbH& Co. KG, Schwabach, Germany) at 20,000 rpm for 60 s.

The particle size distribution was analyzed using a Malvern Mastersizer S (Malvern Instruments Ltd., Worcestershire, UK). The pump velocity was 2000 rpm, the obscuration was between 10 and 20%, and the refractive index (RI) was 1.45 for the miglyol oil and 1.33 for the water. Emulsions were formulated in duplicates (*n* = 6).

### 2.7. Emulsion Stability

The emulsion stability was investigated for emulsions stabilized by rapeseed precipitates precipitated at all pH and was determined by near-infrared light backscattering techniques with a Turbiscan instrument (Formulaction, Toulouse, France). A protein concentration of 10 mg protein/mL oil with 0.1% benzoic acid was used in the emulsion formulations for the stability tests, based on the screening described in Section 2.6. Emulsions were incubated without dilution in cylindrical glass cells at 37 °C for 3 weeks. The vials were scanned every 10 min for the first 2 h, thereafter every 20 min for 2 days, and thereafter twice a day for a total of 21 days. The emulsion droplet size distributions were analyzed immediately after emulsion formulation and after 3 weeks of incubation in the Turbiscan heating block according to Section 2.6.

### 2.8. Statistical Analysis

The rapeseed protein isolation process was conducted in triplicate for each precipitation pH and all analyses were carried out in at least duplicates (*n* = 6). Statistical analyses were performed on the wet mass, dry matter, protein content, protein concentration, protein recovery yield, zeta potential, and precipitation coefficient using SPSS Statistics 26 (IBM, Armonk, NY, USA). The datasets were normally distributed, and a univariate general model was applied. Tukey’s test was used as a post hoc test to investigate significant differences (*p* < 0.05). Difference in emulsion droplet size immediately after formulation and after 3 weeks of storage was analyzed using Mann Whitney U-test.

## 3. Result and Discussion

### 3.1. Extraction

In the first step of the protein isolation process, the extraction step, 52 ± 2% of the proteins were solubilized into the light liquid phase in the alkaline environment (pH 10.5) after 4 h of incubation. In general, the extraction coefficient is highly dependent on the extraction pH, where higher pH values are associated with higher yields. Zhang et al. reported that the extraction coefficient varied between 33% and 62% in the pH interval of 8–13 [36]. Ghodsvali reported an extraction coefficient of 42–45% when using pH 10.5 and around 60% when using pH 12 in the extraction phase [37]. A study performed by Fetzer et al. also investigated the impact of different alkaline pH on the extraction coefficient [15]. Similar to the present study, Fetzer et al. used cold-pressed RPC, although the raw material was defatted with isohexane prior to protein extraction. Their results showed that the highest protein extraction coefficient was found at pH 12 (59.5%), which is a slightly higher extraction coefficient compared with the present study (52%). However, in addition to the higher extraction pH, the addition of NaCl was used during the extraction step, which could also have an impact on the extraction coefficient. However, the raw material is prone to oxidation under extreme alkaline conditions, such as pH 12, with a dark color and bitter flavor developing as consequence. Therefore, extraction at pH 10.5 was chosen as a compromise between the yield and sensorial attributes in the present study.

### 3.2. Precipitation

The wet mass was defined as the total mass including moisture in the precipitate fraction at the end of the process and the wet mass was dependent on the precipitation pH. The largest wet mass of the precipitate fraction was found at pH 3.0 (34 ± 6 g) and the lowest mass at pH 6.5 (14 ± 0 g) (Table 1). The dry matter content followed the same trend, with a lower dry matter at high pH, and an increased dry matter at low pH, although no significant differences were found for pH 3.0, 3.5, and 4.0. The protein content in the precipitate fraction was highest at pH 3.5 and 4.0 (4.7 ± 0.1 g and 4.7 ± 0.0 g) and thereafter decreased as a function of the increased precipitation pH. The protein concentration of the precipitate fraction was highest for pH 4.5 (67 ± 0%, *p* < 0.05), indicating less co-precipitation of non-nitrogenous substances. The precipitation coefficient describes how large the proportion of the solubilized proteins in the light liquid phase obtained in the precipitate fraction after the second separation step is. At lower precipitation pH (3.5–4.5), most of the proteins were detected in the precipitate fraction, with a maximum at pH 3.5 and 4.0 (64 ± 3% and 62 ± 3%, Table 1). At higher precipitation pH, a larger fraction of the proteins remained solubilized and were obtained in the light phase with unprecipitated proteins (pH 5.5–6.5). At pH 6.5, only 32 ± 1% of the proteins were found in the precipitate.

The results are in line with Zhang et al., who found that the precipitation coefficient for cold-pressed rapeseed protein was 59% at pH 3.5 and 62% at pH 4.0, with a significant difference [36]. However, Zhang et al. observed that the maximum precipitation coefficient was at pH 4.5, which was slightly higher than the present study. This is possibly due to the different extraction pH, as Zhang and colleagues used pH 9 instead of pH 10.5 in the present study.

The protein recovery yield describes how large the proportion of the proteins in the starting material, the rapeseed press cake, obtained in the precipitated fraction is. High protein recovery yields correlated with low precipitation pH, with a maximum around pH 3.5–4.0 (33 ± 0%, Figure 2), and at even lower pH, the yield again decreased. The protein recovery yield is correlated to both the extraction pH and precipitation pH [38]. This is because the two process steps are linked: only the protein that was solubilized in the first step of the process can be precipitated in the second step since the non-solubilized protein is left behind in the spent solids fraction. In a study by Gerzhova et al., 3 different extraction pH (pH 10–12) were investigated and the protein recovery yield was 25, 35, and 55% [39], which is in agreement with the present study. Akbari and Wu reported a yield of 51% when rapeseed protein was extracted at pH 12 and precipitated at pH 4–4.5 [26].

### 3.3. Zeta Potential

The zeta potential of the light liquid phase precipitated at different pH was evaluated and the zeta potential was 0 at pH 4.2 (Figure 3), indicating that this should be the pI for the rapeseed protein composition used in the present study. The results agree with the protein recovery yield, with an optimum pH of 3.5–4.0 (Table 1). In this pH interval, the proteins’ surface has a neutral net charge, and the proteins cannot repel each other by electrostatic repulsion. The proteins, therefore, associate in larger clusters and are subjected to forced sedimentation during centrifugation. The results thus indicate that the pI for the protein mixture of the used material is around 4.0. This result is in line with a study performed by Gerzhova et al., where the pI of hot-pressed canola meal was evaluated, and the zeta potential was found to be 0 around pH 4.3 [39]. Moreover, this result is in line with other studies reporting that the lowest solubility for rapeseed protein is in the range of pH 3.7–4.0 [40]. A close botanical relative to rapeseed, *Brassica carinata*, was reported to have the lowest protein solubility between pH 3.5 and 5 [41].

### 3.4. Protein Profile Analysis

To understand how acidic pH influenced the distribution of protein species in the unprecipitated fraction and the precipitate, respectively, the protein profile was analyzed by the lab-on-chip method. A protein profile was analyzed for both the light liquid phase (before acidic pH adjustment) and the resulting supernatants with unprecipitated protein after precipitation at pH 3.0–6.5. Unfortunately, none of the reducing agents achieved clear separation at pH 3.0 and 3.5, probably due to the highly acidic conditions. The proteins in the light liquid phase separated into six different peaks (labeled 1–6 in Figure 4A). The molecular mass of the proteins represented in the first peak was 18 kDa, which corresponds to the size of the oil body protein oleosin (18 kDa) [42]. This peak was present in the light liquid phase and represented 16% of the total proteins (Figure 4B). In the light phase with unprecipitated proteins, the first peak was found at pH 5.5, 6.0, and 6.5 and this peak only represented 1% of the total proteins (Figure 4J–N). At lower pH, no peak was detected, and it could be concluded that the majority of the oleosin protein represented by the first peak was precipitated at all evaluated pH.

The second peak (21–23 kDa) was detected in both the light liquid phase (Figure 4A,B) and the corresponding light phase with unprecipitated proteins at all precipitation pH (Figure 4C–N). The molecular mass of the second peaks lines up with the weight of the β-chain from cruciferin (20–23 kDa) [23]. The protein distribution (Figure 4A,B) shows that the second peak represented 22% of all proteins in the light liquid phase, but for light phases with unprecipitated proteins at precipitation pH 4.5–6.5, this particular peak was estimated to be around 34% (Figure 4E–N). The intensity of this peak was the highest for the light phase with unprecipitated proteins at pH 4.0, with 43% of the total protein in the light phase with unprecipitated proteins (Figure 4C,D). This indicates that the proteins with subunits with a molecular weight of 21–23 kDa had a higher solubility than average in the light liquid phase and did not precipitate to the same extent as other proteins.

The same trend of an increased peak in the light phase with unprecipitated proteins compared to the light liquid phase was followed for the third (27–28 kDa) peak. The molecular mass represented by this peak is a possible match with the molecular weight of the oil body protein caleosin (27 kDa) [13]. This peak represented 25% of all proteins in the light liquid phase (Figure 4A,B) and around 29% in the light phase with unprecipitated proteins at pH 4.0, 5.5, 6.0, and 6.5 (Figure 4C,D,G–N). For pH 4.5, this peak was slightly higher in the light phase with unprecipitated proteins: 34% (Figure 4E,F).

The protein represented by the fourth peak (34 kDa) lines up with the α-chain of cruciferin (29–33 kDa) [23] and represented 24% of all proteins in the light liquid phase (Figure 4A,B). For pH 4.0 and 4.5, it can be seen that the peak height was reduced, with only 18% of all proteins at pH 4.0 (Figure 4D). This indicates that a large amount of the proteins represented in the fourth peak precipitated at these pH. At the remaining pH, the peak was somewhat higher in the light phase with unprecipitated proteins compared to the light liquid phase. The second and fourth peaks also support the findings from a study performed by Qu et al., where SDS-page was used to characterize the profile for rapeseed proteins. Their result showed that the α and β-chains of cruciferin were in the same molecular weight range [43].

When comparing the fifth peak, most of the proteins represented by this peak had precipitated at pH 4.0 and 4.5 (Figure 4C–F). Steroleosin, with a molecular weight of 39–41 kDa, lines up in size with the proteins represented by this peak [13]. For the remaining pH, the protein distribution was similar before and after precipitation, indicating that the solubility of this particular protein was not affected by acidic pH over or below pH 4.0–4.5.

The molecular weight of the last detected peak (53–54 kDa), number six, could be a subunit of cruciferin (50 kDa) [22]. It is possible that the reducing conditions were not sufficient to split all subunits into the α- and β-chains of cruciferin, thus some intact subunits of cruciferin remained in the sample. This also supports the findings from Qu et al., who reported that the cruciferin subunit is in the same weight range [43]. This peak is lower for all light phases with unprecipitated proteins compared to the light liquid phase, indicating that most of the proteins represented by the sixth peak were precipitated at all evaluated pH.

Other peaks presented on the left side of the dotted line were too small to be detected by the chosen protein kit and proteins with a lower molecular weight than 13 kDa were therefore not detected and are not included in the results. Since all analyses were performed under reducing conditions, napin should have split into its 2 chains, 4 and 9 kDa, which falls below the detection limit of the kit used [25]. It can be seen in all graphs (Figure 4A–G) that one or two peaks are visible close to the dotted line. It is possible that these were the peaks represented by the two napin chains. This assumption is in line with a study performed by Lorentz et al. (2014), where rapeseed protein was analyzed using the Bioanalyzer method. They used another protein kit and could thereby observe proteins with smaller molecular weights, such as napin [44].

To understand the composition of the protein in the precipitate fractions, the change in the protein distribution before and after precipitation was analyzed. In the case of a decreased concentration of a given peak, these proteins were enriched in the precipitate fraction and the opposite applied for increased concentrations: these proteins were present in the precipitate to a lesser extent. To summarize, it was found that proteins represented by peaks 1 (oleosin or native napin proteins) and 6 (cruciferin) precipitated to the same extent independent of the precipitation pH. For other peaks, there were differences depending on the precipitation pH. The concentration of proteins represented by peaks 2 and 3 (a beta chain of cruciferin and caleosin) increased in the precipitate at pH in the higher range (pH 5–6.5) whereas a decrease in these proteins was found for precipitates at pH 4–4.5. The opposite was found for proteins represented by peak 4 and 5 (cruciferin and steroleosin), which was enriched in the precipitate at low pH (pH 4–4.5). Precipitated fractions from pH 5–6.5 thereby had a higher concentration of proteins represented by peaks 2 and 3 and a lower concentration of protein represented by peaks 4 and 5 compared to precipitates at pH 4–4.5.

### 3.5. Emulsion Properties

All rapeseed precipitates stabilized emulsions, and the droplet size was dependent on the protein concentration in the formulation, where higher protein concentrations stabilized smaller emulsion droplets (Figure 5). More protein molecules can cover a larger oil–water interfacial area, resulting in smaller droplet sizes. The proteins reach an equilibrium state between the concentration of non-adsorbed protein in the continuous phase and adsorbed proteins in the dispersed phase [4]. At 10 mg protein/mL oil, a plateau was reached for most of the samples, whereafter no further reduction in the droplet size was detected, indicating that the equilibrium between adsorbed and non-absorbed proteins was reached. The droplet sizes (*d*_43_) at this protein concentration ranged from 21–49 µm. Soy lecithin stabilized emulsion droplets with a *d*_43_ of 28 µm at 10 mg protein/mL oil in the emulsion formulation, which falls within the emulsion droplet size interval for rapeseed protein in the present study. This indicated that rapeseed protein can have both higher and lower efficacy as emulsifiers compared to soy lecithin and that the emulsifying properties were, at least to some extent, dependent on the precipitation pH. Rapeseed protein precipitated at low pH (pH 3.0 and 3.5), and soy lecithin did not show this plateau, and the droplet size was further reduced when the protein concentration was increased to 50 mg protein/mL oil. Rapeseed protein precipitated at higher pH (pH 6.0 and 6.5) stabilized emulsions with smaller droplets (*d*_43_ of 21 and 24 µm), indicating that this composition of protein species had higher emulsifying abilities compared to the protein composition in protein powders precipitated at lower pH (such as pH 3.0 and 3.5), which stabilized droplets with a diameter of 37–49 µm (Figure 5). Similar results were found in a previous study in our lab focusing on emulsion formulation and storage conditions [45]. From the protein profile (Section 3.3), it was shown that the oil body protein oleosin was precipitated to the same extent at all acidic pH, whereas the precipitation of other protein fractions increased at lower pH. This means that the purity of oleosin was highest in precipitates at pH 6–6.5. Oleosin is known to be an efficient emulsifier due to its unique structure, with a long hydrophobic region that has the ability to penetrate an oil droplet. We, therefore, suggest that the reason why rapeseed protein precipitated at pH 6–6.5 stabilized smaller droplets than proteins precipitated at lower pH is the enrichment of oleosin. Extreme acidic conditions have been reported to induce the unfolding of protein due to electrostatic repulsion between charged amino acids. These conditions can also lead to dissociation of subunits, breakage of disulfide bonds, reduced molecular flexibility, and an altered balance between exposed hydrophobic and hydrophilic groups, which is an additional obstacle for rapeseed protein precipitated at extreme acidic pH [46]. The emulsion droplet size was overall stable over three weeks in accelerated storage conditions (37 °C) (Figure 6). However, emulsions stabilized by rapeseed protein precipitated at pH 3.0, 4.0, and 6.5 showed an increased droplet size after incubation (Figure 6). The most stable emulsions were stabilized by rapeseed protein precipitated at pH 4.5–6.0. The characteristics of these specific precipitate fractions were the higher concentrations of caleosin and a β-chain from cruciferin and the lower concentrations of steroleosin and an α-chain of cruciferin. From the results, caleosin may be a better emulsifying agent compared to steroleosin, although both are oil body proteins. It is evident that the protein composition has a major role in the stabilization of emulsions and that this composition can be monitored by precipitation pH, although more studies are needed to fully understand protein fractionation on the molecular level.

The present study confirms the findings of Dong et al., who reported that rapeseed protein precipitated at pH 5.8 stabilized emulsions with higher emulsifying capacity and higher oil adsorption ability than protein precipitated at pH 3.6 [30].

Zhang et al. reported that the emulsifying activity and stability of rapeseed protein was maximum when precipitated at pH 3 [36]. When protein was precipitated at higher pH, the emulsifying activity and stability decreased. The different outcome is possibly due to both the different extraction conditions and emulsion formulations. Zhang et al. used pH 9 as the extraction pH and used 0.1% (*w*/*v*) protein in the emulsion formulation. This is a lower protein concentration than the present study, which investigated protein concentrations of 0.5–25% (*w*/*v*) and found that a plateau, after which no further reduction in droplet size occurred, was reached at 5% *w*/*v* (or 10 mg protein/mL oil).

### 3.6. Emulsion Stability

The emulsion stability was monitored for 21 days at 37.5 °C in a Turbiscan instrument. All samples provided homogenous emulsions when they were fresh. However, after a few hours (1–2 h), a clarification layer at the bottom part of the cell appeared for all samples due to the migration of oil droplets to the top part of the cell.

After a few days, a clarification layer also appeared at the top part of the cell in some of the samples, indicating the presence of free oil. This was attributed to an irreversible destabilization phenomenon as a consequence of the oil droplets’ coalescence.

Figure 7A presents a schematic illustration of the samples after 21 days of storage. A larger clarification layer due to the free oil at the top part of the cell was observed for emulsions stabilized by rapeseed protein precipitated at low pH (from 3.0 to 4.5) and pH 6.5. The precipitation pH from 5.0 to 6.0 showed the largest stability and the thinnest layer of free oil on the top part of the cell. The presence of free oil also agrees with the larger particle size measured in Turbiscan after 3 weeks of storage as presented in Figure 6.

The Turbiscan Stability Index (TSI) of the middle part of the cell was also calculated to compare the stability of the different emulsions against the coalescence or Oswald ripening typically found in the middle part (Figure 7B). The TSI value sums all the variations detected in the samples in terms of the size and/or concentration, and is defined by the following Equation (4):(4)$$\mathrm{TSI}=\sum_{i}\frac{\sum_{i}\left|{\mathrm{scan}}_{i}-{\mathrm{scan}}_{i-1}\right|}{H}$$

In a previous study, rapeseed proteins were precipitated at pH 3 and 6 and then used to stabilize emulsions at several pH [45]. These emulsions also showed droplet coalescence when stabilizing emulsions at pH 4.5 at an incubation temperature of 30 °C, which agrees with the present study, where the emulsions had a pH of 4.8.

The surface tension measurements also indicate that the protein precipitated at intermediate pH of 4.5–6 and had lower surface tension, which supports its increased ability to stabilize O/W emulsions (Appendix A).

## 4. Conclusions

This study showed that the precipitation pH affects the yield and protein concentration in the protein precipitate fraction recovered from industrially cold-pressed rapeseed press cake. The highest yield (33%) was found at pH 3.5–4.0, which coincided with the pH at which the rapeseed proteins had the lowest solubility (pH 4.2). The highest protein concentration (67% on dry basis) was found in precipitate fractions precipitated at pH 4.5. The oil body protein oleosin precipitated to the same extent independent of pH, whereas a beta chain of cruciferin and caleosin was enriched in rapeseed precipitates at pH 5–6.5. At lower precipitation pH, there was an enrichment of an alpha chain of cruciferin together with steroleosin. The emulsifying properties were affected by the precipitation pH, and emulsions stabilized by rapeseed precipitates produced at pH 5–6.5 had the smallest droplet size, with even smaller droplets than soy lecithin. After 3 weeks of incubation at 37 °C, the emulsions stabilized by rapeseed proteins precipitated at pH 4.5–6.0 had a similar droplet size as before the onset of incubation and presented significant stability against coalescence or Oswald ripening phenomena. The results show that pH during the isolation process is important for the emulsifying properties and that the emulsion stability is enhanced when precipitation pH of 4.5–6.0 are applied. Since the protein recovery yield is not optimal during this interval, a bio-refinery approach could be evaluated in further studies. By first collecting proteins at a higher precipitation pH (pH 4.5–6.0) and thereafter lowering the pH and collecting the rest, a higher yield can possibly be reached. The two fractions can be used for different purposes, i.e., emulsifying agents and protein concentrates in bulk for food applications.

## Figures and Tables

**Figure 1 molecules-27-02957-f001:**
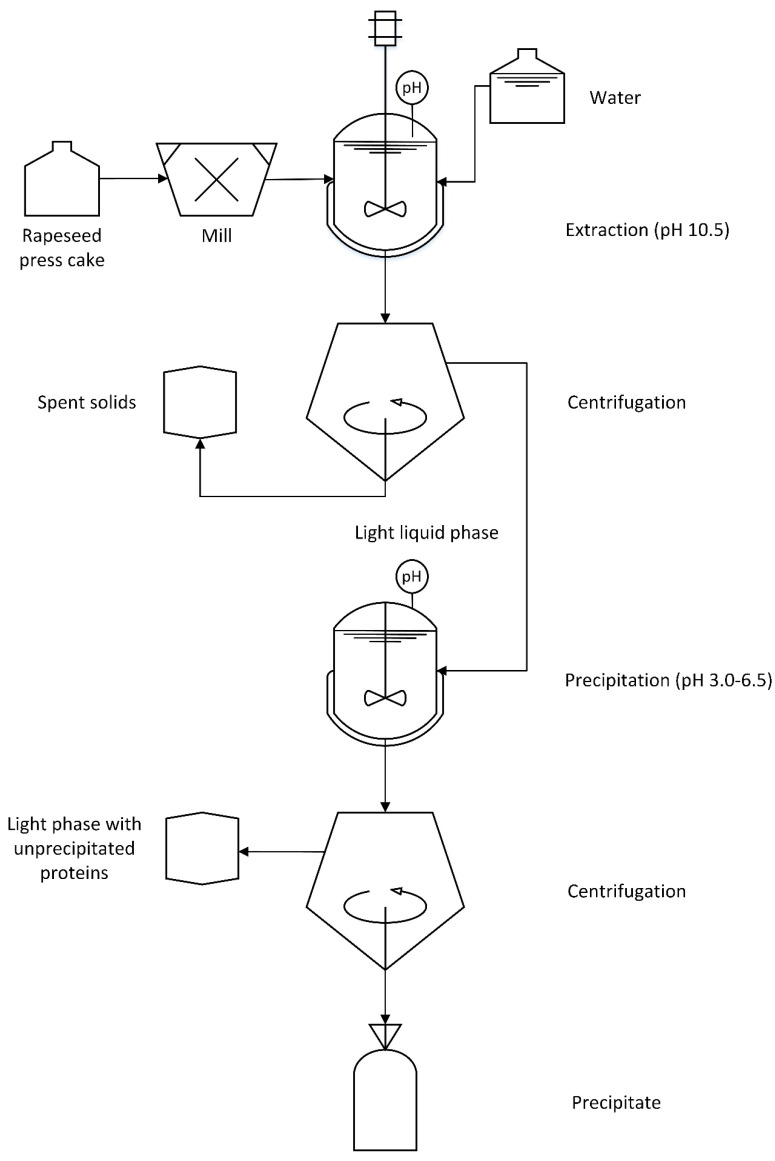
A flowchart illustrating the rapeseed protein isolation process.

**Figure 2 molecules-27-02957-f002:**
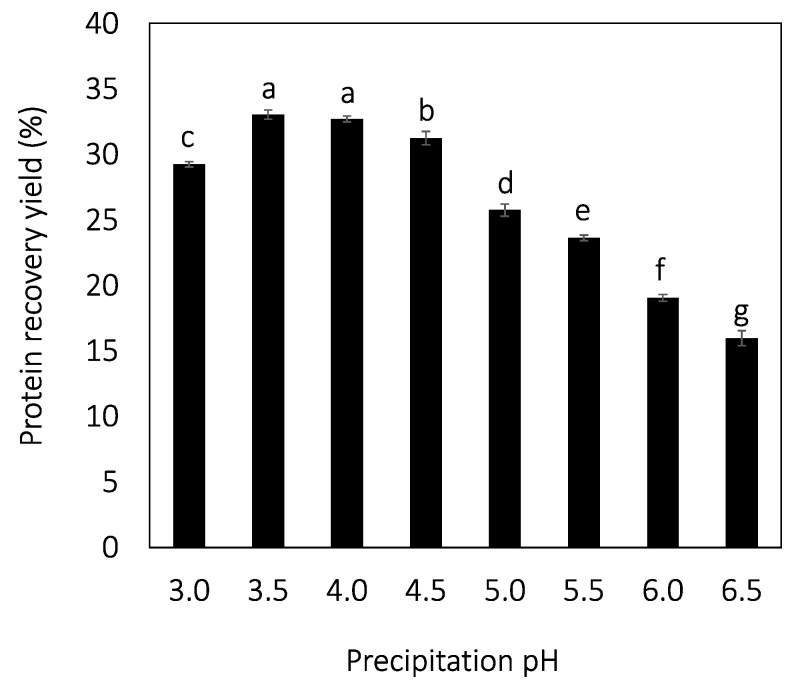
Protein recovery yield as a function of the precipitation pH. Different letters indicate significant differences, *p* < 0.05.

**Figure 3 molecules-27-02957-f003:**
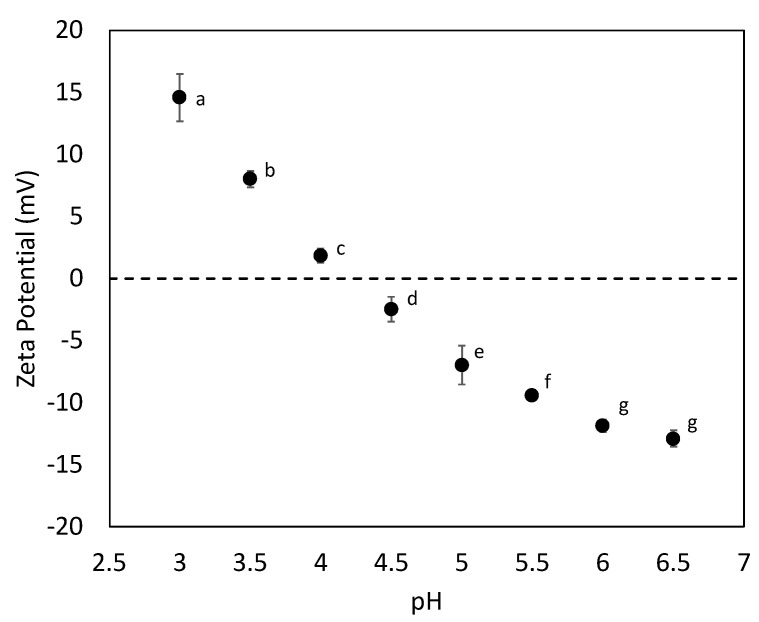
The zeta potential of the light liquid phase precipitated at different pH. Different letters indicate significant differences, *p* < 0.05.

**Figure 4 molecules-27-02957-f004:**
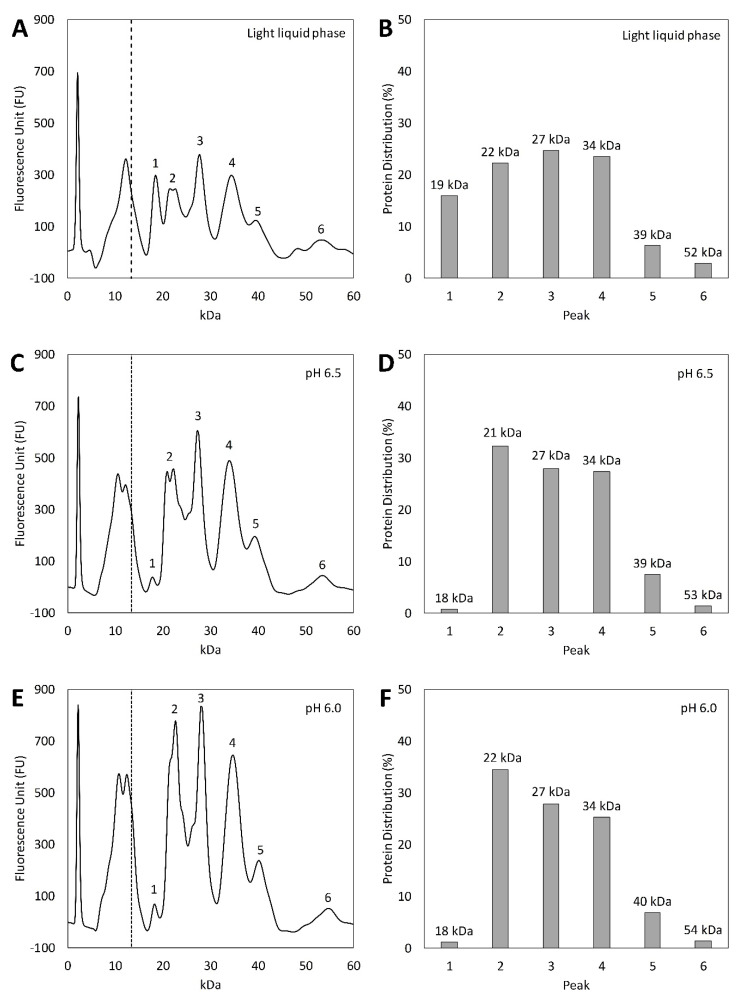
Protein profile and protein distribution are sorted by the six different peaks and their molecular weight span. Figure (**A**,**B**) represents the light liquid phase and (**C**–**N**) the light phase with unprecipitated proteins at various precipitation pH. The dotted line at 14 kDa represents the detection limit of the method and a method-specific system peak is displayed at 4.5 kDa.

**Figure 5 molecules-27-02957-f005:**
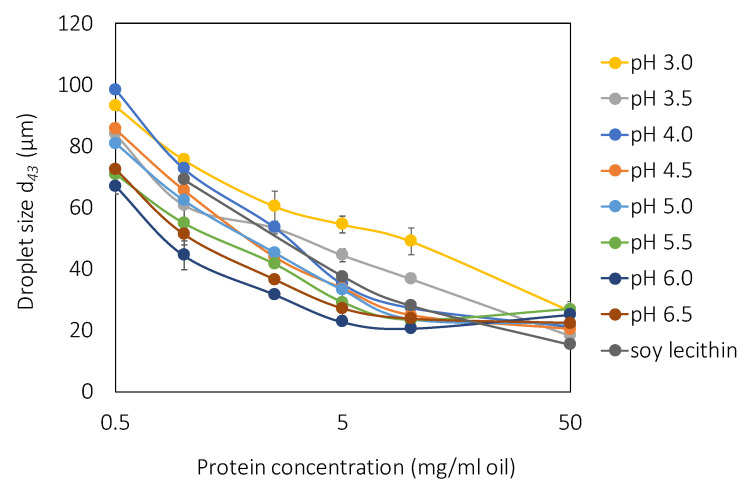
Emulsion droplet size (*d*_43_) as a function of the protein concentration in the emulsion formulation with soy lecithin as the control.

**Figure 6 molecules-27-02957-f006:**
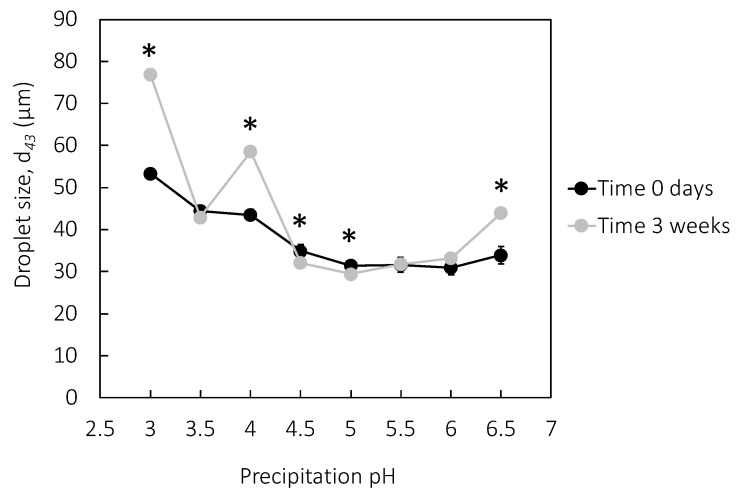
Emulsion droplet size immediately after formulation and after 3 weeks of storage at 30 °C. the emulsifier concentrations in the emulsions were 10 mg protein/mL oil. Star indicates significant differences in droplet size between time 0 and after incubation for 3 weeks, *p* < 0.05.

**Figure 7 molecules-27-02957-f007:**
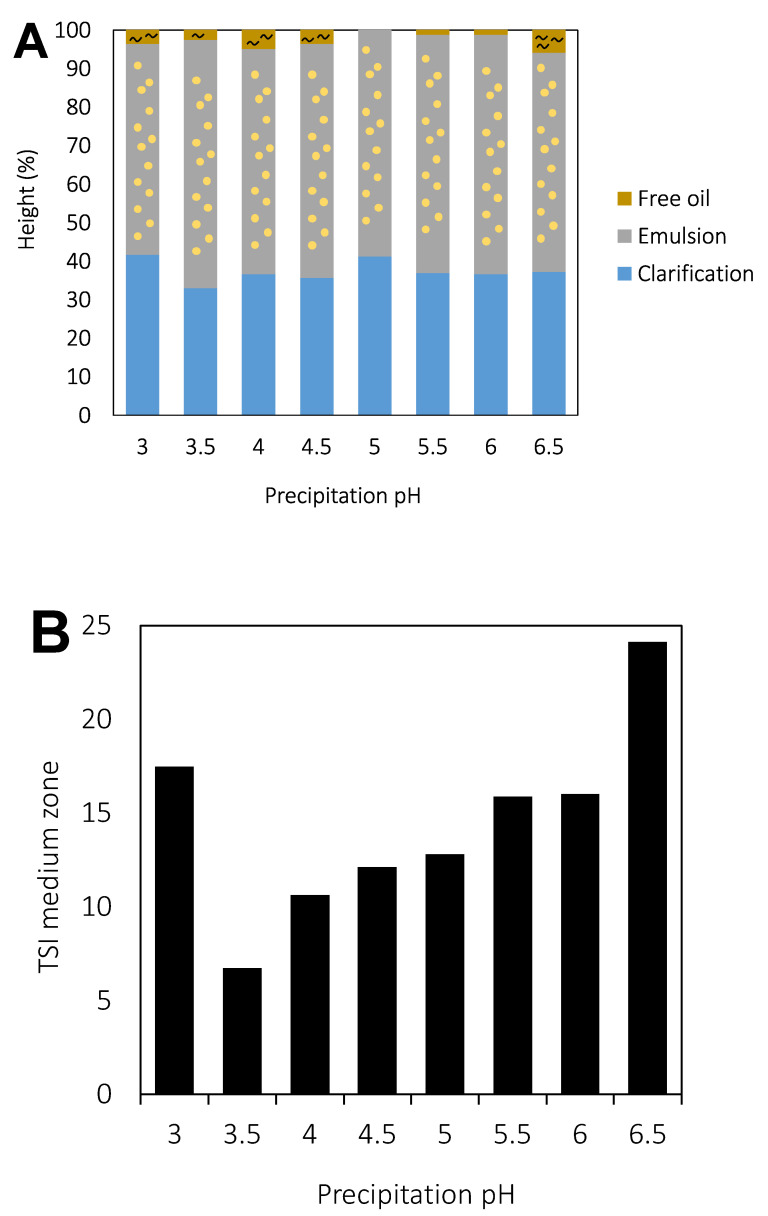
(**A**) Schematic diagram of the phases observed for emulsions prepared at different pH after 21 days (30 °C) of storage. (**B**) Turbiscan Stability Index (TDI) of the middle part of the cell for the emulsions prepared at different pH.

**Table 1 molecules-27-02957-t001:** Wet mass, dry mass, protein content and concentration, and protein recovery yield as a result of the precipitation pH. All data are expressed as means ± standard deviation. Different letters in each row indicate significant differences, *p* < 0.05.

Precipitate Fraction								
	pH 3.0	pH 3.5	pH 4.0	pH 4.5	pH 5.0	pH 5.5	pH 6.0	pH 6.5
Wet mass (g)	34 ± 5 ^a^	29 ± 1 ^b^	25 ± 0 ^bc^	21 ± 0 ^cd^	18 ± 0 ^de^	19 ± 1 ^de^	16 ± 0 ^e^	14 ± 0 ^e^
Dry matter (g)	7.4 ± 0.4 ^a^	7.6 ± 0.1 ^a^	7.3 ± 0.1 ^a^	6.7 ± 0.1 ^b^	5.7 ± 0.1 ^c^	5.5 ± 0.1 ^c^	4.6 ± 0.0 ^d^	4.0 ± 0.1 ^f^
Protein (g)	4.2 ± 0.0 ^c^	4.7 ± 0.1 ^a^	4.7 ± 0.0 ^a^	4.5 ± 0.1 ^b^	3.7 ± 0.1 ^d^	3.4 ± 0.0 ^e^	2.7 ± 0.0 ^f^	2.3 ± 0.1 ^g^
Protein concentration dry basis (%)	56 ± 3 ^g^	62 ± 1 ^c^	64 ± 1 ^b^	67 ± 0 ^a^	64 ± 1 ^b^	61 ± 1 ^d^	59 ± 1 ^e^	57 ± 3 ^f^
Protein precipitation coefficient (%)	53 ± 0 ^c^	64 ± 0 ^a^	62 ± 0 ^ab^	59 ± 1 ^b^	52 ± 0 ^c^	46 ± 0 ^d^	37 ± 0 ^e^	32 ± 1 ^f^

## Data Availability

The datasets generated for this study are available on request to the corresponding author.

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
