# Peer review of "The Effect of Precipitation pH on Protein Recovery Yield and Emulsifying Properties in the Extraction of Protein from Cold-Pressed Rapeseed Press Cake"

_molecules, 2022, doi:10.3390/molecules27092957_

Round 1

Reviewer 1 Report

In this paper, the authors reported the effect of precipitation pH on protein recovery yield and the emulsifying properties in the extraction of protein from cold pressed rapeseed press cake. The paper fit the aims and scope of Molecules. I would recommend accepting the paper after modifications.

  1. In my opinion, the abstract need to be improved by providing more information about the article.
  2. Introduction should be substantially improved to clarify the novelty of the manuscript. The author should justify the value of the work and compare their work with previously similar published papers. The introduction section needs to be elaborated. The high-value utilization of by-products in agro-food industry can increase economic value and environmental benefits and better highlight sustainability. The author should comment on studies on the extraction of high-value target substances from by-products or waste and related applications. Therefore, the incipit has to be supported with proper suitable literatures

doi: 10.1016/j.lwt.2021.111617, doi:10.3390/foods9040449, doi: 10.1016/j.ijbiomac.2018.02.018, doi: 10.1002/jsfa.11055, doi:10.3390/polym13132044

  1. It was strongly suggested to indicate at the end of the Introduction section the main employed characterisation techniques in order to achieve their purpose.
  2. Please check the abscissa of Figure 5.
  3. It might be better to discuss the emulsifying properties in conjunction with protein composition and droplet size.
  4. It might be better to improve the characterization of the emulsion system. The following literature can be used as a reference

doi: j.foodhyd.2021.107180, doi: 10.1016/j.lwt.2022.113337

Author Response

Reviewer number 1 – round #1

Q1: In my opinion, the abstract need to be improved by providing more information about the article.

Response: Thank you for your valuable comment. We have now added more results in the abstract such as zeta potential, extraction coefficient and turbiscan data (lines 18-26).

Q2: Introduction should be substantially improved to clarify the novelty of the manuscript. The author should justify the value of the work and compare their work with previously similar published papers. The introduction section needs to be elaborated. The high-value utilization of by-products in agro-food industry can increase economic value and environmental benefits and better highlight sustainability. The author should comment on studies on the extraction of high-value target substances from by-products or waste and related applications. Therefore, the incipit has to be supported with proper suitable literatures.

doi: 10.1016/j.lwt.2021.111617, doi:10.3390/foods9040449, doi: 10.1016/j.ijbiomac.2018.02.018, doi: 10.1002/jsfa.11055, doi:10.3390/polym13132044

Response: Thank you for pointing this out. We agree and have now updated the introduction. We have highlighted the novelty (lines 105-122) and added more information about other studies working with high-value target substances from by-products (lines 34-44). We have also added information about other studies concerning extraction efficiency (lines 67-76).

Q3: It was strongly suggested to indicate at the end of the Introduction section the main employed characterisation techniques in order to achieve their purpose.

Response: Thank you. We have now added more information about the characterization techniques at the end of the introduction section, lines 118-122.

Q4: Please check the abscissa of Figure 5.

Response: Thank you for pointing this out. We have now changed the absicca in Figure 5 to 0.5-50 protein concentration (mg/ml oil).

Q5: It might be better to discuss the emulsifying properties in conjunction with protein composition and droplet size.

Response: We agree that emulsifying properties should be linked to protein composition and droplet size. We have added droplet size data to the result section to make it easier for the reader to follow (lines 438-439, 448 and 451) and have also added a comparison with soy lecithin (lines 440-444). Emulsifying properties such as droplet size and protein composition are discussed on lines 458-460 and lines 467-472.

Q6: It might be better to improve the characterization of the emulsion system. The following literature can be used as a reference doi: j.foodhyd.2021.107180, doi: 10.1016/j.lwt.2022.113337.

Response: We thank the reviewer for the suggested literature and have added one of the references (lines 433-436) which provided arguments for why it is important to find the critical concentration of emulsifiers in a system. We will expand our emulsion characterization protocol to also include e.g. confocal laser scanning microscopy which provided an excellent view of droplet shape and packing pattern, oil volume fraction and oil binding ability for our next study. Also, the analysis of secondary structure (Yang et al) was interesting and we will have this in mind for the next study. In our previous studies, we have already investigated lipid oxidation and antioxidant capacity of different botanical varieties of rapeseed in the emulsion context and we agree that these are important matters (doi:10.3390/foods9050678 and doi:10.3390/foods8120627). Emulsions stabilized by rapeseed protein were less prone to oxidation compared to both BSA and soy lecithin which was related to the protein-lipid interaction in the rapeseed botanical matrix.

Reviewer 2 Report

The article is preapred correctly. Research consers investigation perticipation pH affects protein yield, protein content and emulsifying properties with industrially cold pressed rapeseed press cake as starting material.

The results show that isolation process and emulsion stability depend on pH applied. The limited number of studies show impact of pH on protein isolation. If the rapeseed press cake, rich in high-quality proteins with could become a new plant protein food source, extraction process should be examined.

The introduction is correctly written and informative, fully revealing the research aspect.The extraction method is well described.

The results are presented legibly, the discussion is carried out correctly. The authors describe the extraction method in detail, the article is prepared correctly. The number "n" remains to be clarified, because a too low number "n" may reduce the scientific value of the article.

My comments for Authors are presented below.

The materials and methods are described correctly, but I cannot conclude how many samples were analyzed in the experiment. What was the number "n"?

Please explain the aspect that appears in the lines:

  • 150, I don't understand the sentences: "The analysis was performed in at least duplicates."
  • 157, 164,similarly here; "....at least duplicates"
  • 180, What does the " at least triplicates" mean? 
  • 198, Please explain: " ...at least duplicate and each emulsion was measured two times and the average was reported."
  • Table 1 line: 242, In the description to the table, please, replace "mean" with means
  • line: 265, 266, "statistical difference" please, replace with significant
  • Figure 3 Joining the points in the graph recommends a continuity that is not here. The points on the graph should not be connected.

line:149, 157, 164

What does the last sentence mean, were there only two samples? What was the number n?

Author Response

Reviewer number 2 – round #1

Q1:
The materials and methods are described correctly, but I cannot conclude how many samples were analyzed in the experiment. What was the number "n"?

Response: The rapeseed protein isolation process was performed in three batches per precipitation pH, which means that a total of 24 batches were prepared. For each batch, the analysis was performed in duplicate or triplicate. This means that the protein content for protein precipitated at pH 6.5 was measured 6 times, pH 6.0 was measured 6 times, pH 5.5 was measured 6 times etc. For the Bioanalyzer, the equipment has a larger degree of variation and therefore each batch was measured 3 times which means a total of 9 measurements for each precipitation pH.  This has been clarified on the following line 174, line 186, line 193, line 199, line 216, line 235 and lines 250-251.

Q2: Please explain the aspect that appears in the lines:

  • 150, I don't understand the sentences: "The analysis was performed in at least duplicates."
  • 157, 164, similarly here; "....at least duplicates"
  • 180, What does the " at least triplicates" mean?
  • 198, Please explain: " ...at least duplicate and each emulsion was measured two times and the average was reported."

Response: Thank you for pointing this out. The isolation process was performed in triplicate and each analysis was performed in at least duplicate. The number n is thereby 6 for all analyses except for the Bioanalyzer (n=9), which has been clarified on the following line 174, line 186, line 193, line 199, line 216, line 235 and lines 250-251.

Q3: Table 1 line: 242, In the description to the table, please, replace "mean" with means

Response: Thank you for this comment. We have now replaced mean with means on line 281.

Q4: line: 265, 266, "statistical difference" please, replace with significant

Response: Thank you for this valuable comment, we have now replaced statistical differences with significant on lines 304-305.

Q5: Figure 3 Joining the points in the graph recommends a continuity that is not here. The points on the graph should not be connected.

Response: Thank you for pointing this out. We agree and have deleted the line connecting the data points in figure 5.

Q6: line:149, 157, 164. What does the last sentence mean, were there only two samples? What was the number n?

Response: Thank you for this comment. See responses to comments 1 and 2 above for a more detailed explanation.

Round 2

Reviewer 1 Report

The revised manuscript has been improved and some evidence proposed by Authors can be accepted.